# Bird song comparison using deep learning trained from avian perceptual judgments

**Lies Zandberg** [1,2]*, **Veronica Morfi**[3], **Julia M. George**[2,4], **David F. Clayton**[2,5], **Dan Stowell**[3,6,7], **Robert F. Lachlan**[1,2]

**1** Department of Psychology, Royal Holloway University of London, Egham, United Kingdom, **2** Department of Psychology, Queen Mary University of London, London, United Kingdom, **3** Machine Listening Lab, Centre for Digital Music (C4DM), Queen Mary University of London, London, United Kingdom, **4** Department of Biological Sciences, Clemson University, Clemson, South Carolina, United States of America, **5** Department of Genetics and Biochemistry, Clemson University, Clemson, South Carolina, United States of America, **6** Department of Cognitive Science and AI, Tilburg University, Tilburg, Netherlands, **7** Naturalis Biodiversity Centre, Leiden, Netherlands

* elisabeth.zandberg@rhul.ac.uk

**Data Availability Statement:** All data and code is available on a GitHub repository at https://github.com/veronicamorfi/ml4bl. We have also used Zenodo to assign a DOI to the code: 10.5281/

## Abstract

Our understanding of bird song, a model system for animal communication and the neurobiology of learning, depends critically on making reliable, validated comparisons between the complex multidimensional syllables that are used in songs. However, most assessments of song similarity are based on human inspection of spectrograms, or computational methods developed from human intuitions. Using a novel automated operant conditioning system, we collected a large corpus of zebra finches' (*Taeniopygia guttata*) decisions about song syllable similarity. We use this dataset to compare and externally validate similarity algorithms in widely-used publicly available software (Raven, Sound Analysis Pro, Luscinia). Although these methods all perform better than chance, they do not closely emulate the avian assessments. We then introduce a novel deep learning method that can produce perceptual similarity judgements trained on such avian decisions. We find that this new method outperforms the established methods in accuracy and more closely approaches the avian assessments. Inconsistent (hence ambiguous) decisions are a common occurrence in animal behavioural data; we show that a modification of the deep learning training that accommodates these leads to the strongest performance. We argue this approach is the best way to validate methods to compare song similarity, that our dataset can be used to validate novel methods, and that the general approach can easily be extended to other species.

## Author summary

How do birds hear the differences between their songs? This fascinating question carries implications, since the study of bird song, a model system for the neurobiology of learning and animal communication, depends critically on our ability to assess the similarity of songs. Traditionally, researchers compare sounds by human assessment, or use computational methods based on human intuitions about similarity. However, neither approach is connected to birds' own perception of sound similarity. Here, using a novel automated

zenodo.5545932; and the data: 10.5281/zenodo.5545872.

**Funding:** This research was supported by Biotechnology and Biological Sciences Research Council research Grant. No. BB/R008736/1 "Machine Learning for Bird Song Learning." awarded to RFL, DS and DFC. https://www.ukri.org/councils/bbsrc/ The funder did not play a role in the study design, data collection and analysis, decision to publish, or preparation of the manuscript.

operant conditioning system, we recorded many thousands of acoustic judgments of similarity from zebra finches, and used this perceptual decision data for the first time to train a deep learning system. The trained system outperforms other computational methods for the task of making the same judgments as birds. This algorithm to compare song similarity, together with the potential of extending the general approach to other species, places the study of bird song on a firmer footing.

## Introduction

Bird song is an important model system for several related fields: the neurobiology of learning, animal communication, sexual selection and cultural evolution [1–3]. Similarities in development, neurobiology and neurogenomics have further led song to become a model system for understanding human speech [3, 4]. Its importance largely results from the fact that songbirds memorise songs they hear from adult conspecifics during a sensitive phase early in life, and then produce imitations of these songs [1], but this flexibility also leads to bird song being an unusually variable animal signal.

All these fields of bird song research depend on reliable comparisons of song recordings. This task is not straightforward because bird songs, like many acoustic signals, are very high dimensional—with many potential spectral features varying dynamically through the course of each song unit. In fact, when animals themselves compare two vocal signals, they must subjectively integrate numerous differences in frequency and timing. This leads to the realisation that the comparison methods we use need to be validated against animals' own perception: there is no comparison method that can be objectively "correct" [4, 5].

The first studies comparing bird song were performed by comparing songs by ear or transcribing songs into onomatopoetic descriptions [6]. The invention of the sonograph revolutionized the study of animal communication [7], making it possible to produce a visual representation of the time-frequency structure of the songs on the basis of which spectrograms of songs can be compared visually: human visual assessment of spectrographic similarity (HVA). To assess similarity more repeatably, Clark et al. [8] proposed spectrographic cross-correlation (SPCC) as a computational measure of similarity. Two spectrograms are superimposed and then shifted temporally to find the peak correlation coefficient, which is used as a measure of similarity between the sounds. A recent implementation of this method can be found in software such as Raven Pro [9]. SPCC relies heavily on the spectrographic representation of the signal, for example being intolerant of often minor differences in the relative duration of spectral components. An alternative approach is to extract contours of acoustic features that are believed to be perceptually relevant, and then to align the contours of two signals and integrate the differences between them in these features. Sound Analysis Pro (SAP) is a widely used tool that employs this approach, measuring several features (e.g. Wiener entropy, spectral continuity, pitch and frequency modulation), that can be tuned to the study species [10]. SAP allows for timing differences by linearly warping time by up to 30%. Another software package, Luscinia [11], also uses multiple acoustic features to generate a dissimilarity measure, but uses dynamic time warping (DTW) to align features. DTW allows non-linear warping of time to find the optimal alignment of features. SAP and Luscinia both require decisions about which acoustic features to use and how to weight them.

Recently, data-driven methods have been proposed for the analysis of song similarity based on machine learning [12–14]. While not yet widely used in bird song research, such methods reduce the dependence on engineering good features, by using spectrograms or waveforms

directly as input. Deep learning can achieve impressive accuracy on various tasks, but this neither implies nor demands that their recognition strategies are similar to those of animals [15]. All comparison methods make assumptions about what constitutes similarity in acoustic signals. This is true even in the case of unsupervised deep learning, which (like principal components analysis before it) derives a representation automatically from unlabelled data: in such a case, assumptions about similarity emerge implicitly from the choice of training data as well as the neural net structure.

The most fundamental problem facing all comparison methods is how their assumptions are validated [5]. The design of the algorithms relies on the intuition of the scientists, although particular decisions can be justified on the basis of perceptual research. But, typically, they have only been validated by comparison with the previous "gold standard", HVA. However, HVA is based on human, not avian, perception, in the visual, rather than auditory domain, of a spectrographic representation of a sound, which differs in known ways from how sounds are perceived (e.g. the linear rather than logarithmic scaling of frequency [16]).

To overcome these problems, in the present work we have: 1) developed a novel method to test zebra finches for their perception of sound similarity and deployed it to collect a large data-set of avian sound assessments, 2) developed a novel algorithm to train a deep neural network to assess song similarity, and 3) validated this algorithm as well as current conventional methods of measuring song similarity on basis of the birds' assessments. By using an automated operant feeder in combination with radio-frequency identification (RFID) technology we were able to individually train and test group-living zebra finches continuously, enabling us to collect over 900,000 responses to stimuli. Such response data typically involve some amount of inconsistent or unclear responses, so we further adapted our algorithm to train using both the unambiguous and the ambiguous operant response data. Although here we tested birds for their assessments of similarity in zebra finch syllables, this method can be applied to measure similarity between arbitrary new song unit recordings. Unlike previous methods, this algorithm is not validated by HVA, but by a measure of the birds' own perceptions of sound similarity. Here we found that the algorithm we developed on the basis of the birds' assessments outperformed all widely used methods.

## Materials and methods

### Assessments of song similarity

**Ethics statement.** Animal procedures were approved by the UK Home Office and conducted under UK Home Office Procedures Project License P3B2BBD6B.

**Birds.** We experimentally tested group-living zebra finches for their perception of sound similarity using a novel operant conditioning system. Experiments were performed using 26 domesticated zebra finches from an outbred colony at Queen Mary University of London. All birds used in the experiment were hatched in the population between December 2015 and May 2018 and reared in a flock in a large free-flight room. The birds were fitted with a RFID (radio frequency identification) tag leg band (Eccel Technology Ltd., Leicester, UK) in addition to the standard metal leg ring and a colour band. Throughout the experiment all birds were kept at a 12.00:12.00 light:dark schedule (lights on at 7:00 GMT), in two separate aviary rooms (one room with two aviaries of approx. 100x200x200cm and one room with two aviaries of approx. 200x200x200cm). Each experimental aviary held 4–6 individuals in both single-sex and mixed groups. The birds received water *ad libitum*, and a commercial tropical seed mixture from the operant feeders. Each aviary was outfitted with 2 operant feeders. Birds could feed *ad libitum* for a half hour after lights on, and a half hour before lights off. Operant feeders were operational from 7:30 to 18:30, which meant that the birds had to interact with the feeder

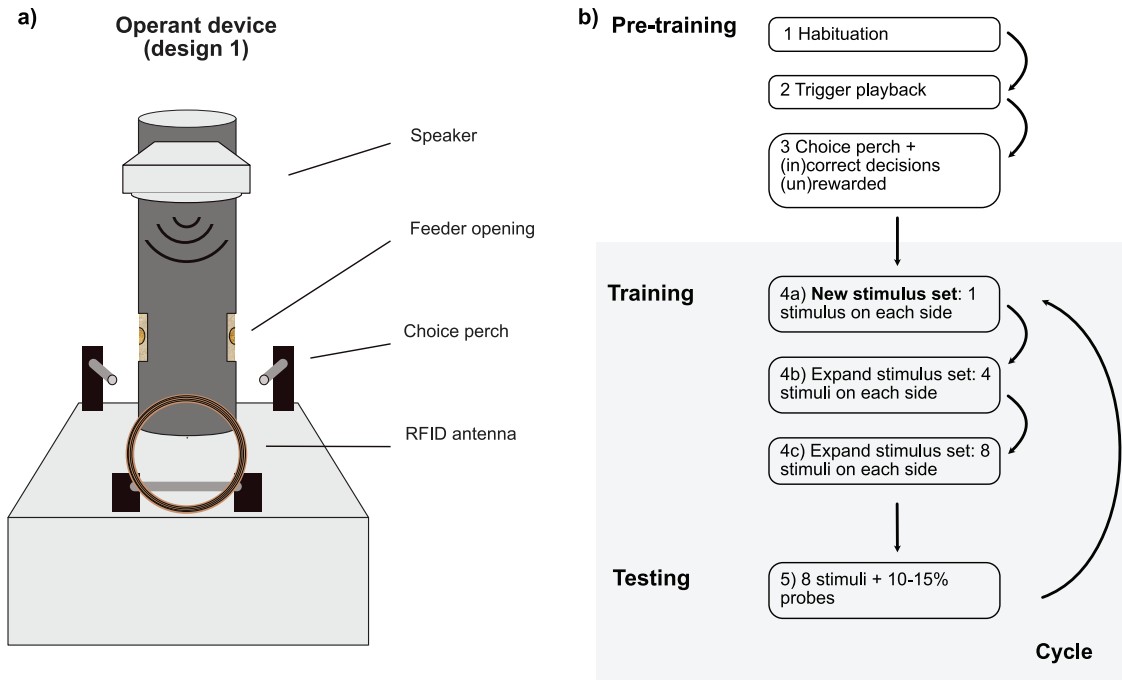

**Fig 1. Schematic image of operant feeder and overview of the bird training scheme.** (A) Schematic image of operant feeder design 1. (B) Bird training scheme with the different training steps. During pre-training birds were habituated to the device, and learned how to operate the device to obtain a food reward. After pre-training finished, they proceeded with their first testing cycle. Each testing cycle (step 4a—5) consisted of a training period in which the birds were trained with new sets of stimuli (step 4a-4c), followed by a testing period (step 5) during which each bird is still presented with the same set of 8 training stimuli, but now interspersed by 10–15% probes. Once a sufficient number of probes had been collected, the testing period, and with it the current cycle, was finished. The birds subsequently started a new cycle by starting training with a new stimulus set (step 4a).

to gain access to the food. Every day the performance of each bird and its number of successful visits was monitored. Birds that did not interact with the apparatus or were consistently unsuccessful were removed from the experiment. This occurred in two cases: one bird did not interact with the device, and one bird developed an unrelated health problem during the experiment. Additionally, birds were weighed before the experiment started and after 2 training cycles to check for excessive changes in weight.

**Operant device.** We used two different operant feeder designs (see Fig 1a for a schematic image of design 1, and S1 Fig for a photo of the operant feeder). Each feeder device consisted of a PIT tag detection system, a motor-activated feeding tube, two capacitive touch perches registering responses, speaker system, a Raspberry Pi 3B+ and a Hall effect sensor. In design 1 all electronics are integrated in a weatherproof electronics box, whereas in design 2 these components are integrated in drainpipe elements. Additionally, in design 2 the stimuli are played through two speakers, whereas in design 1 there is only one speaker. Other than the external casing and the number of speakers the devices work identically. The Raspberry Pi computer receives information from the PIT tag system and the response perches, and on the basis of this input controls the speaker output and the running of the motor activating the reward mechanism. A PIT antenna was positioned at 20cm from the feeding tube, which registered the bird passing through. On opposite sides of the feeding tube two aluminium response perches were placed, which registered the presence of a bird on the perch through a capacitive touch sensor. The feeding mechanism consisted of a PVC outer tube with an opening on both the left and the right side, and inside this tube a transparent acrylic inner tube with only one

opening. This inner tube held all the seeds used as a reward. By turning the inner tube so that the opening lined up with the opening in the outer tube, birds were able to access the seeds in the inner tube. To prevent misalignment of the inner tube a Hall effect sensor was built in together with a magnet attached to the inner tube, to check the position of the inner tube after each reward, and to readjust when needed.

**Sound stimuli.** Sound stimuli used were 887 syllables extracted from zebra finch song recordings [17]. These song recordings were made in a different population from the focal population to avoid any potential confounding effects in case of recognition of singer identity. All songs were recorded at 48kHz and divided into separate syllables, which were high pass filtered (100Hz), normalised and a 20ms fade in and out was added. When a stimulus was played it was repeated 4 times in quick succession with a 20ms interstimulus interval.

**Test paradigm.** We tested group-living captive zebra finches for their perception of sound similarity using an operant feeder with a 2 alternative forced choice (2AFC) paradigm: forced-choice 'AXB' judgments (Fig 1A). In short, this meant that we trained birds to discriminate between two sound stimuli, 'A' and 'B' and associate each of them with one of the choice perches. Once the birds were accurate in their responses to these training stimuli we introduced 10–15% probe stimuli 'X', which in terms of similarity lies somewhere in between the two sets of training stimuli, for them to assess for their perception of similarity. When a bird was presented with a probe stimulus we expected it to respond by choosing the perch of the training stimulus that it perceived to be most similar to the presented probe. We infer that the choice made in response to the AXB (probe-stimulus) triplet represents a perception of categorical similarity. A trial was started by a bird passing through the PIT antenna and registering its identity on basis of the PIT tag. Depending on the identity of the bird, the software selected a stimulus sound to be played through the speaker(s) on the top of the device. Each stimulus sound is either a training stimulus, and thus associated with a 'correct' side of the device, left or right, or is a probe sound. Responses to training sounds were rewarded only when the bird responded by hopping onto the perch associated with that side. In case of an incorrect response, a short fragment of white noise was played and the feeder remained closed. All responses to probe sounds were rewarded by the inner feeding tube opening, through which the birds had access to the seeds for 2 seconds. For a more detailed description of the steps involved in pre-training, training and probe presentation see below and Fig 1B. Since training and testing were done in a group setting birds could learn from, or scrounge from others' actions. Scrounging occurred either by displacing the original bird to claim the reward after a successful operation of the device, or by hopping on a choice perch after the focal bird triggered sound playback. Although this may have increased noise in our data, testing in a group setting made it possible to train and test continuously for months, without the welfare considerations that come with testing in social isolation.

**Selection of stimuli.** Using the DTW in Luscinia sound software [11] we compared all syllables with each other and extracted dissimilarity measures for each combination of syllables. Based on these dissimilarity measures, we subsequently constructed a matrix with, for each syllable, a ranking of all other syllables ranging from the most similar to the least similar syllable. From all syllables a random syllable was chosen as the first training stimulus, stimulus 'A', and a second training stimulus, stimulus 'B', was selected to be ranked between 50 and 150 similar to the first training stimulus. For each stimulus A and B, their 7 most similar syllables were selected as additional stimuli. Probe stimuli were selected to be ranked between 20 and 200 of both training stimuli (see S2 Fig).

**Pre-training.** *Step 1*—Habituation to the feeder: Feeder opening moves from left to right side, every 30 minutes. Birds can feed *ad libitum* from the feeder. *Step 2*—Birds have to go through the antenna, which triggers a playback of a stimulus. After the playback the feeder

immediately opens to the side of the feeder associated with the sound. The feeder remains open for 8 seconds during which the bird can feed. *Step 3*—The time delay to open after playback is increased. When a bird hops onto a choice perch within the time delay period it receives either a correct (stimulus is repeated and the feeder opens immediately) or an incorrect response (playback of 3s white noise, and an added time delay of 30s before the next trial can be initiated). If the bird does not make a choice within the time delay period the feeder opens automatically after the time delay. Step-by-step the time delay period is increased to encourage the birds to make a choice, rather than to wait for the feeder to open automatically. At the same time, the reward time of the automatic opening is reduced from 8 seconds to 3 seconds, whereas the choice-reward time is kept at 8 seconds.

**Training.** *Step 4a*—The time delay to open automatically is removed and from this phase onward the feeder does not open automatically anymore: the birds have to make a correct choice to receive a food reward and the the choice-reward time is reduced to 2 seconds. Similar to the previous phase an incorrect response by the bird is followed by a playback of 3s white noise, and an added time delay of 30s before the next trial can be initiated. In this training phase each bird is trained with its own set of stimuli: one stimulus is rewarded on the left side, and the other stimulus is rewarded on the right side. *Step 4b*—When on average the group makes 70% accurate decision we increased the number of stimuli on each side to 4. These stimuli are selected to be the original stimuli plus the 3 syllables most similar to each of the original stimuli. Cycle accuracy was calculated as the proportion of correct responses among all of a bird's responses to the training stimuli over the total of a cycle's test period. *Step 4c*—When on average the group makes 70% accurate decision we increased to 8 stimuli on each side (original stimulus and the 7 most similar stimuli).

**Probe testing.** *Step 5*—After the group reaches on average 70% accurate decisions with 8 stimuli on each side we added 10–15% probes. The responses to probe sounds are always rewarded. For each probe testing period, we aimed to collect approximately 1000 probe decisions in total (all birds together). After reaching this number the current cycle was finished and the birds would start a new cycle by starting training with a new stimulus set (Fig 1B).

## Deep learning embeddings

We next built a deep learning model that can learn from the birds' judgements about similarity of sounds and use it to create a perceptual space that can mirror birds' perception. This perceptual space is a learnt vector representation of data referred to as an *embedding* space and can be used for classification, verification and other similarity-based tasks [14, 18, 19].

Our deep learning approach follows that of [20], but is adapted specifically to animal behavioural data, which can, due to the limited accuracy and consistency of measured animal judgments, contain a high proportion of uncertain data points. We define ambiguous data points to mean those for which, even after general data quality filtering, we do not have a consensus in the birds' decisions for a given stimulus.

In common with [21], our method is distinct from most deep learning embedding methods (including triplet-based methods) in that no semantic "classification" labels are used. It is also distinct from purely unsupervised methods which are guided only by the signal variation in the dataset, and thus have no necessary link to perceptual or communicative relevance [13].

In order to infer a data-driven similarity algorithm from our probe responses, it was required to fit a highly non-linear function to the data, where the algorithm input is the audio data (transformed to power mel spectrograms), and the fit is conditioned on the responses to our probes. Deep learning has made great advances in creating learnt *embeddings* by methods

such as triplet networks which rely on distance metrics rather than classification as the training objective [22–25].

Triplet networks are trained on triplets of data points consisting of an anchor, a positive sample, and a negative sample. Through the training procedure an embedding space is learnt so that the anchor and the positive samples are featured close to each other, while at the same time the anchor and negative samples are separated as much as possible. However, most previous successes in representation learning are driven by datasets with explicit class labels, which are used to provide a strong signal of semantic distance even for triplet networks [26]. In other words, the distance metric is often derived from an underlying measure of classification correctness, rather than general similarity.

Our operant AXB experiments were designed so a probe X can correspond to a triplet anchor. Based on the A/B decision a bird made for that probe, all training stimuli rewarded on the selected side correspond to positive exemplars for that anchor, while all training stimuli rewarded on the opposite side correspond to negative exemplars in the triplet. This makes it possible for us to create a dataset of triplets out of the decisions made by the zebra finches in order to train a triplet network to learn an embedding space for the recordings used in the experiment. We are able to then project new stimuli into this learnt embedding space to measure their similarity to the training stimuli. This is done without any use of labels or class knowledge for the stimuli. The goal is for the learnt embedding space to make the same A/B decisions as birds would make for the recordings.

**Network architecture.** An overview of the deep learning model architecture for processing a single stimulus is depicted in S3 Fig. We compute the log-scaled power mel spectrogram from the zebra finch recordings, of 150 mel bands and 170 time frames, and use it as input to our model. The selection of log mel spectrogram as input was based on prior work which found these useful for deep learning applied to birdsong data [20](cf. [27] for a comparison between different sound representations of zebra finch vocalisations). The parameters for the Mel spectrogram computation are set as follows: the length of fast Fourier transform (FFT) window equals 2048, the number of samples between successive frames is set to 128 with window length 512. The architecture of our model consists of a shared convolutional part followed by two parallel parts of attention pooling and max pooling, the predictions of which are combined at the last layers and projected to an embedding space of $d$ dimensions (see [20] for more information).

For our triplet model, the anchor, positive and negative examples are each passed separately through the network depicted in S3 Fig, to estimate the coordinates of each stimulus in the perceptual embedding space. The distances between these coordinates are then used as input to the triplet loss function, described next, which will be used to update the weights of the network.

We reiterate that, in common with [21], our method is distinct from most deep learning embedding methods (including triplet-based methods) in that no semantic "classification" labels are used. Our algorithm training is driven by the data coming from two-alternative similarity decisions, using both the ambiguous and unambiguous behavioural data.

**Metric learning with triplet loss.** Let $f(x) \in \mathbb{R}^d$ represent an embedding that maps a sample $x$ (an audio clip) into a $d$-dimensional space. Often $f(x)$ is normalised to have unit length for training stability [28]. We want to ensure that an anchor sample $x_a$ is closer to positive samples $x_p$ that are more perceptually similar to it than negative samples $x_n$ without having any information about the class of each sample. We define the distance between any two points $x_i$ and $x_j$ as $D_{ij}$. In our experiments, this distance denotes the Euclidean distance. For dataset of

N stimulus triplets $\mathcal{T} = \{x_a^i, x_p^i, x_n^i\}_{i=1}^N$ of example stimuli the loss is given by

$$\mathcal{L}(\mathcal{T}) = \sum_{i=1}^N [D_{ap}^i - D_{an}^i + \delta]_+ \qquad (1)$$

where $[\cdot]_+$ is standard hinge loss and $\delta$ is a non-negative margin hyper-parameter. The larger the margin, the further the negative example should be from the anchor in the embedding space [19, 25, 26].

**Unambiguous triplet setting.** In order to build a perceptual space that reflects zebra finch perception, we require a set of easily distinguishable signals to avoid perceptual ambiguity. Bird decisions for sound similarity can contain a lot of noise; this can be due to the cycle accuracy of the individual, scrounging interactions or the consistency of probe decisions caused by perceptual ambiguity of sounds. In triplet learning it can be important to create a dataset without any ambiguity in it, and in our experiments we try to limit this noise from the above sources as much as possible.

In order to reduce decision noise and make our dataset unambiguous we perform the following pre-processing steps on the bird decision data:

- Discard individual's decisions if cycle accuracy for them is less than 65% on the training stimuli. (We use training stimuli to estimate this, since we cannot calculate it from the probes: we do not make any assumptions about their positive/negative association with the training stimuli.)

- Discard decisions if there were any technical issues with the device for that day.

- Discard inconsistent probe decisions: a consistent probe decision is one where a bird made at least two decisions for a particular probe during a cycle, and of those decisions, the bird chose the same side in > 70% of the decisions.

This unambiguous dataset of decisions is constituted of 2,099 triplets. In total 547 probe syllables were used as an anchor for these triplets and each triplet was assessed on average 4.326 ± 2.51 times (mean ± sd). We split these into training and evaluation sets of 1,239 and 860 triplets, respectively. The training triplets are used as input to our deep learning model in order for it to learn a space that can satisfy the triplet metric of Eq 1. The evaluation set only consists of triplets acquired from birds with an accuracy of 77% and higher to make it as unambiguous as possible and to properly evaluate the performance of our model.

**Ambiguous triplet setting.** In [21] a method is introduced to learn perceptual embeddings for haptic signals, gathered from human participants. Similarly to our data, theirs was based on two-alternative decisions about texture similarity between triplets of items. Their model is trained on both unambiguous and ambiguous triplets and they show an improvement over training only on unambiguous data.

To make use of the ambiguous data points for training our model, we collected bird decisions that had an inconsistency in side selection. Decisions with 50% to 70% side consistency were chosen from individuals with cycle accuracy of 65% and higher to produce a dataset of "ambiguous" triplets. This ambiguous dataset of decisions totals 1,073 triplets. We split these into training and evaluation sets of 668 and 405 triplets, respectively. Unlike common triplet-based learning approaches which tend to ignore these types of triplets as uninformative, our approach treats both triplet types (ambiguous and unambiguous) as informative, based on the work in [21]. Deep learning benefits from diverse training data. The key intuition is that an ambiguous triplet is not bad data, but is inherently informative: the embedding should arrange points such that ambiguous triplets should not have an unambiguous embedding distance.

Depending on whether a triplet is ambiguous or unambiguous, a different condition needs to be satisfied. For an unambiguous triplet $x_a, x_p, x_n$: $D_{an}^2 - D_{ap}^2 \geq \delta$; on the other hand for an ambiguous triplet $x_a, x_p, x_n$: $D_{an}^2 - D_{ap}^2 = 0$, where $D_{ij}$ is the Euclidean distance between any two points $x_i$ and $x_j$ and $\delta$ is a non-negative margin hyper-parameter. These conditions are defined as:

$$p_u = D_{an}^2 - D_{ap}^2 - \delta \tag{2}$$

$$p_a = D_{an}^2 - D_{ap}^2 \tag{3}$$

In order to facilitate training with both ambiguous and unambiguous triplets, for a triplet $\mathcal{T} = \{x_a^i, x_p^i, x_n^i\}_{i=1}^N$ we adjust the loss function of Eq 1 to:

$$\mathcal{L}(\mathcal{T}) = \sum_{i=1}^N [u(1 - \exp^{p_u}) + (1 - u)(1 - \exp^{-|p_a|})]_+ \tag{4}$$

where $[\cdot]_+$ is standard hinge loss and $u \in [0, 1]$ denotes if a triplet is ambiguous ($u = 0$) or unambiguous ($u = 1$). This loss differs slightly from that of [21], and in pilot studies was found to be more stable for training.

In some of our machine learning tests we use only the unambiguous decisions; we also test the use of ambiguous and unambiguous data together. Furthermore, we explore models that use triplets created by Luscinia after tuning the algorithms parameters using the birds' decisions (we call these *Luscinia-U triplets*, as explained in the following section). In either case the training data are treated as a single training dataset, to train the network by backpropagation using the Adam optimiser, following standard practice in deep learning.

**Evaluating deep learning.** To explore the sensitivity of our deep learning method to hyperparameter settings, we evaluated its performance at different dimensionalities of embedding space, and different ways of using bird decisions and Luscinia-U decisions in training (Table 1). To evaluate whether an algorithm can produce similarity measures compatible with the birds' judgments recorded in the operant devices, we measured the degree to which each algorithm produced the same triplet decision as did the birds. We focussed our evaluation on a held-out set of unambiguous sounds. Due to the inherent variability of behavioural decision data, the gold standard is not 100% but is an accuracy rate matching that of the birds. This rate cannot be measured directly since the ground truth is unknown for the test probes; we estimated it from their cycle accuracy as *estimated maximum attainable accuracy (cycle accuracy)*.

**Table 1. Table of attributes for all learnt embedding space through our deep learning model.**

|  | Emb-U | Emb-LU | Emb-LUA | Emb-Pre | Emb-PreUA | Emb-PreLUA |
|---|---|---|---|---|---|---|
| pre-trained (Pre) | - | - | - | X | X | X |
| Luscinia triplets (L) | - | X | X | - | - | X |
| unambiguous bird decisions (U) | X | X | X | - | X | X |
| ambiguous bird decisions (A) | - | - | X | - | X | X |

In the table, *pre-trained (Pre)* refers to models that used Luscinia-U triplets to pre-train with and initialise the weights of the deep learning model; *Luscinia triplets (L)* refers to models that used Luscinia-U triplets for training (104457 triplets); *unambiguous bird decisions (U)* refers to models that used the triplet decisions created by the bird judgements through our operant experiments (2099 triplets: 1,239 training and 860 evaluation); *ambiguous bird decisions (A)* refers to models that used the triplet decisions created by the bird judgements through our operant experiments (1073 triplets: 668 training and 405 evaluation). Note that some models used more than one source of triplets to train.

Cycle accuracy was calculated as the proportion of correct responses among all responses to training stimuli during the test phases. We also calculated an upper bound based on the birds' within- and between-bird agreement on the test probes, calculated as the percentage of decisions that agree with the majority decision for a given triplet, averaged over all triplets for which there was more than one decision recorded. This is an optimistic upper bound since it relies on a consistency assumption (namely, that each bird's majority decision is the correct decision for a AXB triplet). We refer to it as the *estimated maximum attainable accuracy (consistency assumption)*.

## Luscinia parameter tuning

Separately, we used the bird decisions (from the unambiguous triplets) to tune the parameters of the Luscinia software [11]. We refer to this tuned software as Luscinia-U.

Luscinia uses dynamic time warping (DTW) to align syllables, based on the trajectories of several acoustic features [11]. In this case, these features were: fundamental frequency, peak frequency and mean frequency (log-transformed), fundamental and peak frequency change (the arcsin transform of the slope of these features on the spectrogram), normalized fundamental frequency (where the syllable-wide mean of fundamental frequency is subtracted from the fundamental frequency), Wiener entropy and harmonicity (measures of spectral complexity), and vibrato amplitude (a measure of sinusoidal signal in the fundamental frequency). Finally, time itself is included as a feature in order to penalize time warping. When two syllables are compared using this DTW process, a Euclidean distance is calculated over each of these features for each point in one syllable compared with each point in the other syllable. A dynamic algorithm then searches for an efficient alignment between the two syllables using these distances, and an overall dissimilarity is then calculated by averaging dissimilarities over the alignment.

A key challenge is to decide how to weight the sound features relative to each other. We developed a data-driven approach that used the same unambiguous triplets that we used to train neural networks. For each triplet in the dataset, we calculated the likelihood that the DTW algorithm would make the same choice as the birds, by calculating the dissimilarity between probe and the two training stimuli as: $X_p = (D_{probeA} - D_{probeB})/(D_{probeA} + D_{probeB})$. With this dissimilarity $X_p$ we integrated log-likelihoods over the entire training dataset: $1/(1 + e^{(4X_p)})$, and used a Metropolis-Hastings acceptance rule, to update or keep the weights used in the DTW algorithm. We used flat priors across potential parameter values. This Monte Carlo Markov Chain approach found DTW weightings that would maximise likelihood. The MCMC chain was initiated with equal weightings for each feature. We ran it for 10,000 iterations, and discarded the first 1,000 iterations. New parameter values were sampled from a log-Gaussian distribution with standard deviation 0.1. In each generation, parameters were normalized to sum to 1. We then estimated the mean parameter weightings from the last 9,000 generations of the MCMC and used those in a DTW analysis to measure the dissimilarity between each pair of syllables in the dataset. This dissimilarity matrix was then used for further analysis and comparison of different methods. The weightings of acoustic features before and after training are shown in S1 Table.

We evaluated Luscinia-U directly, but we also experimented with using triplet decisions from that system as additional training data for deep learning, to supplement the bird decisions dataset. For this, we used the tuned Luscinia-U to generate additional AXB triplet decisions. These were then used as additional data, either to pre-train the deep learning network or simply pooled with the natural bird decisions during the main training phase. For this we

filtered the Luscinia-U triplets down to only the coarse-level ('easy') distinctions using a minimum distance threshold, to minimise the risk of contradicting birds' judgments.

## Other software

We compared the performance of our deep learning methods with current software used for similarity measures: Raven Pro [9], Sound Analysis Pro [10] and Luscinia [11] (with and without tuning). For each software package, we extracted the (dis)similarity matrix of the evaluation set and used this to determine their AXB decisions. From this we calculated their accuracy at predicting bird AXB decisions.

## Acoustic features used by the algorithm

It is hypothetically possible that our trained deep learning algorithm would rely entirely on just one or a few basic underlying acoustic features, such as fundamental frequency or frequency change. To test this, we carried out MRM (Multiple Regression on Distance Matrices) [29] analyses to understand which acoustic features best predicted the 64-dimension embedding of our algorithm. Since this embedding was trained on birds' perceptual judgements, this analysis also indirectly reflects birds' perceptual weightings of different features.

We computed a Euclidean distance matrix between all syllables in our data-set, based on their embedding in 64-dimensional space. We then extracted summary statistics (mean, maximum, minimum, start and end) for each of five acoustic features (fundamental frequency, peak frequency, fundamental frequency change, Wiener entropy, harmonicity) plus syllable length. Syllable length and fundamental and peak frequency were log-transformed. We then scaled each of the 26 measures by subtracting their mean and dividing by their standard deviation.

Since we expected that our comparison method is highly multidimensional, we first examined the importance of all 25 summary statistics (5 for each of the 5 acoustic features), and syllable length. We ran an MRM analysis (permutations = 10,000) to explore how these 26 predictor matrices predicted syllable dissimilarity according to our trained algorithm (Emb-LUA).

Subsequently, to get a better understanding of which *overall* features the birds pay attention to, we calculated, for each acoustic feature, one dissimilarity matrix that combined the 5 summary statistics, using the distance function in the R package *ecodist* (version 2.0.10) [30]. We then ran an MRM analysis (permutations = 10,000) to explore how these 6 predictor matrices predicted syllable dissimilarity according to our trained algorithm.

Finally, we examined to what part of the syllable the birds pay attention to by calculating distance matrices for the mean, start and end summary statistics, and ran a third MRM analysis.

## Results

### Training and test outcomes

Using RFID technology we were able to continuously train and test each bird individually within its home cage and social group, enabling us to collect sound assessments over long time periods and on a much larger scale than conventional operant experiments. We trained 23 birds in 4 aviaries with in total 99 sets of stimuli. Each stimulus set was used once to train one bird. Over a period of 11 months we presented the birds with 1,116,214 trials, for which we received 927,158 responses (totals include all trials in the training phases step 4a-4c, and the testing phase step 5). Of these trials, 25,999 trials were probe trials from which we collected

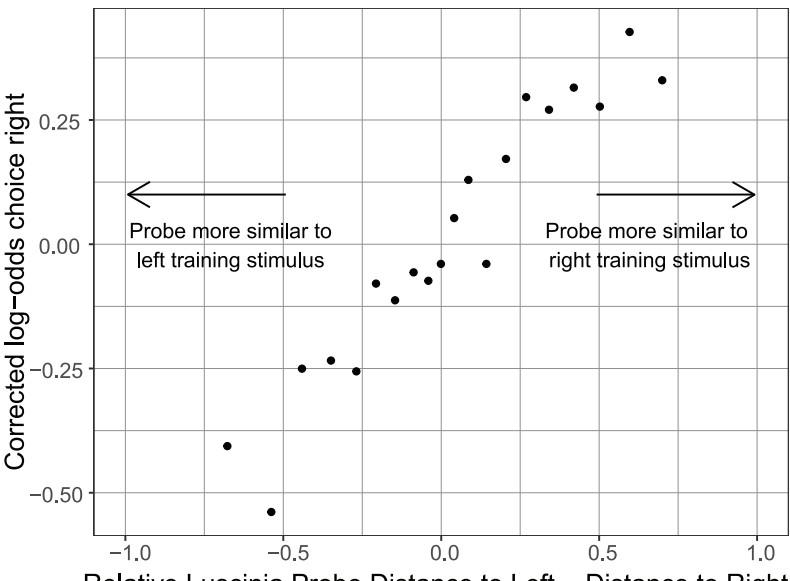

**Fig 2. Dissimilarities between probe and stimuli, according to Luscinia, generally predicted zebra finch choices of perches.** The x-axis plots $(RD_{ProbeLeft} - RD_{ProbeRight})/(RD_{ProbeLeft} + RD_{ProbeRight})$ where $RD_{ProbeLeft}$ is the rank distance of the probe to the left training stimulus and $RD_{ProbeRight}$ is the rank distance of the probe to the right training stimulus. The y-axis shows the log-odds of zebra finches choosing the right perch, adjusting for the side bias of the entire dataset (l.o.=0.584). Trials were binned into 20 equally sized groups based on the relative Luscinia probe distance.

22,048 sound assessments from the birds of in total 753 probe syllables. In total, birds assessed the same probe on average 3.93 ± 2.53 times (*mean ± sd*). For each bird and each testing period, we calculated a cycle accuracy, which is the overall accuracy in its responses to the training stimuli during a specific test period. Out of 99 stimulus sets, for 59 sets the birds reached a cycle accuracy of 65% or higher (mean accuracy of birds performing > 65% ± *sd* = 73.9% ± 5.5%).

To assess whether birds made the choices we expected, we compared the birds' assessments of similarity with assessments made by Luscinia. Dissimilarities between probe and stimuli, according to the Luscinia DTW algorithm, generally predicted zebra finch choices of perches (Fig 2). See also S4 Fig for example spectrograms of a stimulus pair and 4 probes that were assessed by the bird, along with the bird's classification of the probes.

## Deep learning

To explore the sensitivity of our deep learning method to hyperparameter settings, we evaluated its performance at different dimensionalities of embedding space (S5 Fig), and different ways of using bird decisions and Luscinia-U decisions in training (Table 1 and Fig 3). Good performance was obtained with embeddings of 16 dimensions or more, peaking at 64 dimensions (S5 Fig).

We evaluated our algorithms by measuring the degree to which each algorithm produced the same triplet decisions as birds. We found that the deep learning performance is improved by our interventions, especially Emb-LUA—which used the unambiguous and ambiguous triplets as well as the additional Luscinia-U triplets—but not by pre-training (Fig 3). Using a mix of Luscinia-U decisions made on triplets of sounds along with perceptual decisions from the birds provided the best result.

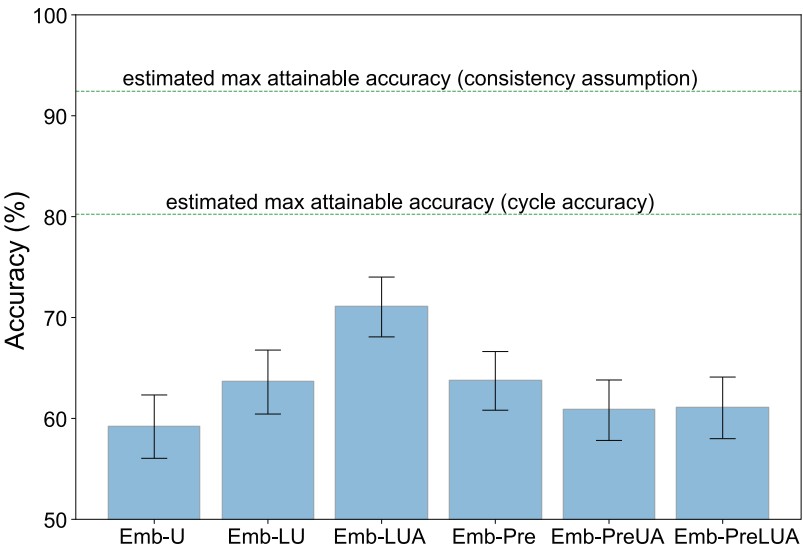

**Fig 3. Deep learning performance is improved by our interventions, but not by pre-training.** Accuracy performance of the different variants of our deep learning model, each model being described in Table 1. Error bars show 95% confidence intervals (bootstrap estimate). The upper lines indicate two different ways to estimate the maximum accuracy attainable in principle; 100% is not attainable since birds' choices are not always consistent for the same stimulus set. An accuracy of 50% represents chance level.

We furthermore investigated the similarities between all models by projecting them in a 2-D space by using multidimensional scaling (MDS) [31] (Fig 4). This analysis does not use acoustic data but only the AXB predictions/decisions as input to MDS: the MDS distance between two algorithms should reflect the number of times they disagree. In this visualisation, algorithms are closer together if they produce similar decisions, irrespective of whether those decisions are right or wrong. It illustrates that all three pre-trained deep learning models remain very similar, and indeed very similar to Luscinia-U, the source of their pre-training, whereas our Emb-LUA model produces a different pattern of judgments.

## Other software

When comparing the performance of our deep learning model with current software used for similarity measures—Raven Pro [9], Sound Analysis Pro [10] and Luscinia [11] (with and without tuning)—we found that deep learning Emb-LU trained using standard unambiguous triplets along with Luscinia-U triplets was unable to outperform these specialist tools (Fig 5). However, our deep learning model Emb-LUA trained on both unambiguous and ambiguous triplets along with Luscinia-U triplets can learn an embedding space that can reflect the birds' perceptual judgements better than any current software publicly available. Comparison between Emb-LU and Emb-LUA leads us to conclude that even AXB triplets that yield inconsistent side decisions can provide useful information about perceived sound similarity, which helps to constrain the optimisation search space for deep learning. The difference in accuracy between the two best performing models, Emb-LUA and Luscinia-U, is 2.9 percentage points. We added all software methods to the MDS (multidimensional scaling) defined 2-D space that enables the comparison of model output (Fig 4).

When comparing software methods, we found that Sound Analysis Pro would yield similarities of precisely zero for some comparisons (in the case of high dissimilarity). In triplets where the Sound Analysis Pro analysis yielded a zero value for the comparison between the probe

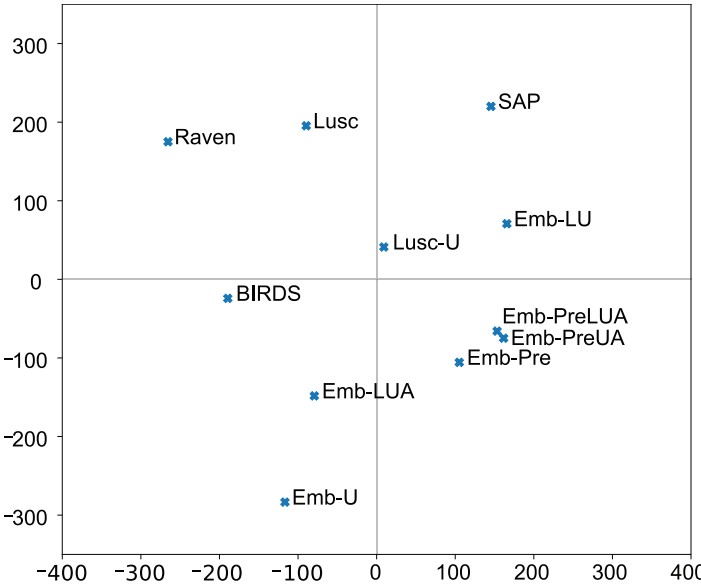

**Fig 4. Different algorithms differ from birds' judgements of similarity in different ways.** 2-D multidimensional scaling (MDS) projection of the similarity between different algorithms based on their predictions. *BIRDS* is the ground truth provided by the bird decisions. Current software methods: *SAP* (Sound Analysis Pro). *Raven*, *Lusc* (Luscinia), *Lusc-U* represents the Luscinia software with tuned Luscinia parameters. Deep learning embedding models: *Emb-** are as described in Table 1.

and both training syllables of the triplet, we assume the three syllables to be so dissimilar to each other that no similarity judgement can be computed from Sound Analysis Pro; this was the case for 103 triplets. To inspect whether these highly dissimilar triplets led to an unfair penalty for some systems, we repeated our evaluation with those 103 triplets excluded (S6 Fig). This did lead to improved performance of Sound Analysis Pro, though the overall pattern of

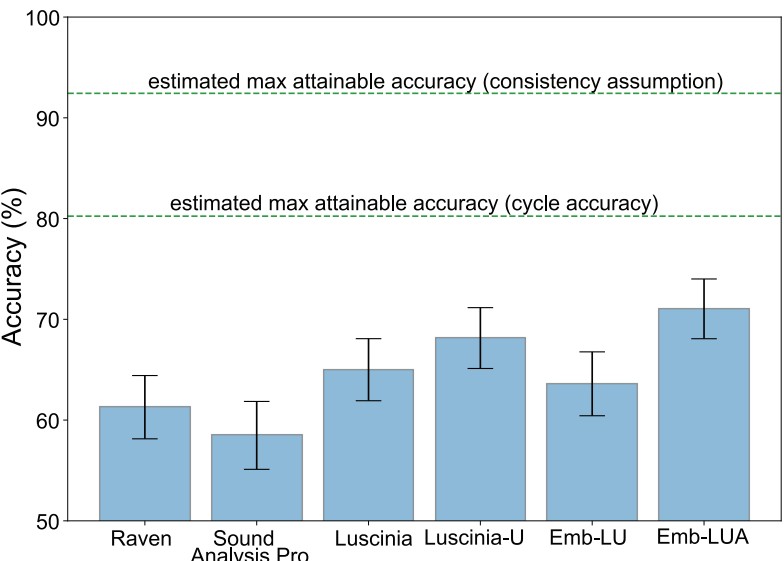

**Fig 5. Accuracy of computational methods at predicting bird AXB decisions, evaluated on the evaluation set.** Details are the same as in Fig 3.

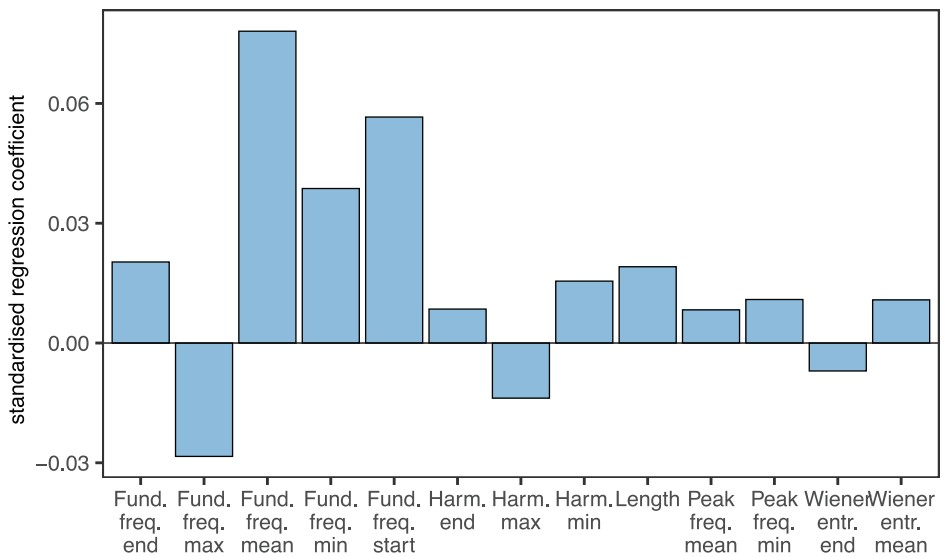

**Fig 6. Outcome of MRM analysis, comparing the Emb-LUA embedding with summary statistics of 5 song features and syllable length as measured in Luscinia.** Figure only shows the standardised regression coefficients for the significant features, for the full model table see S2 Table.

outcomes remained similar, with our Emb-LUA model maintaining the best performance over all other systems.

## Acoustic features used by the algorithm

We carried out 3 different MRM analyses to understand which acoustic features and summary statistics best predicted the 64-dimension embedding of our algorithm (Emb-LUA). For the first model examining all 26 measures, the standardised regression coefficients suggested particularly strong effects for the mean, minimum, maximum and start of the fundamental frequency, and weaker effects for several summary statistics of harmonicity, peak frequency, Wiener entropy as well as syllable length (Fig 6 and for full model results S2 Table). This analysis, which accounted for 68% of overall dissimilarity between syllable embeddings, suggested a wide range of predictor feature/statistic pairs—13 of them were significant and 8 of them had a standardised regression coefficient of >0.01, with the strongest predictor, mean fundamental frequency, scoring 0.078.

For the second analysis, where all measures are grouped by the 5 acoustic features plus syllable length, the standardised regression coefficients suggested particularly strong effects for fundamental frequency, syllable length and, to a lesser extent, harmonicity and peak frequency (S7A Fig). The model as a whole predicted 63% of variation in embedding scores. Finally, when grouping variables by summary statistic, we found that Start, End and Mean all predicted, with similar normalised regression coefficients, the embedding significantly, suggesting that the birds pay attention to all parts of the syllable. The model predicted 37% of the variation in embedding scores (S7B Fig).

These results support the claim that our comparison method makes use of highly multidimensional attributes of birdsong, and, indirectly, that song perception by birds might do too.

## Discussion

Making reliable comparisons of song recordings is essential for the field of bird song research. However, so far most comparison methods use HVA for validation, rather than the birds' perception itself. We compiled a large corpus of zebra finches' perceptual decisions of syllable similarities, which allowed us to validate existing methods, and also to train a deep neural network to assess song similarity.

Training and testing birds in a group setting allowed us to collect a dataset with over 22,000 probe assessments; to our knowledge no similar dataset has been generated and its size allows training of deep learning algorithms as we demonstrate. The social context of training, combined with relying on behavioural rather than neural responses potentially adds additional sources of noise compared with other approaches (e.g. [32]). While in our experiments some birds reached accuracy levels of over 80%, some other birds' performance remained consistently low in accuracy. One reason for these differences in accuracy may be that the group setting allowed some birds to scrounge others' rewards [33], or to interfere with other individuals' trials. In future experiments our method could be refined by adding an extra RFID antenna to the choice perches to limit choice registration to only the bird that started the trial. But even in social isolation, however, similar operant conditioning designs do not lead to birds approaching 100% accuracy: birds continue to frequently explore unrewarded options (e.g. [34]). These sources of error are substantially higher than when human observers are asked to generate perceptual training data-sets.

The birds' choices in operant tests were, however, much more similar to computational methods than expected by chance. This suggests that overall the birds make decisions that correspond to acoustic characteristics, rather than, for instance, choosing randomly or always the same side. The differences between computational assessments and the birds' decisions are difficult to interpret: we might expect that it can be explained partly by the noise in the data due to the birds' behavioural variation in responses, and partly by differences between the algorithm and birds' underlying perception. Taking into account, (1) our large sample size, (2) our inclusion only of birds that performed to a reasonably high level on training choices, and (3) evidence of agreement between trials when making the same choice on probes, we have made the assumption that a higher agreement between birds' choices and computational judgement reflects a greater similarity between bird perception and the algorithm.

To train the birds to discriminate beyond two training syllable exemplars, we needed to ensure that they would generalise their decisions to other syllables too. To achieve this we trained the birds with two sets of 8 very similar syllables (rather than just two exemplar syllables). But this methodological choice created a potential conundrum: we had to make assessments of similarity between these 8 similar syllables using an existing method—in this case, Luscinia. Potentially, this might have influenced how birds learned: providing information about what was regarded as similar or not. At an extreme, training the birds might be the same as training them about the algorithm used to pick the training stimuli. Our protocol, however, involved selecting the 7 most similar syllables to an exemplar, with the rationale that the two categories of training stimuli would be very clearly separable and non-overlapping, no matter what the comparison method. We checked the similarity rankings of these training sets between Luscinia, Raven (SPCC) and Sound Analysis Pro (S8 Fig), and in all cases the two training sets are very clearly separate, showing a higher similarity for the within-set syllables, than the between-set syllables. Therefore, we believe that the structure of our stimulus categories (for this part of the experiment) is providing minimal information to the birds about the nature of similarity.

Using a mix of Luscinia-U decisions made on triplets of sounds along with perceptual decisions from the birds provided the best correspondence to bird decisions (Fig 3). We noticed that a pre-training phase using Luscinia-U triplet decisions led to worse performance. This finding is in contrast to much recent work in deep learning that makes heavy use of pre-training, especially when target datasets are small [35, 36]. It can be attributed to the fact that the pre-trained space (driven by Luscinia's similarity algorithm) is not appropriate, because it initialises the model such that its fine judgments follow Luscinia rather than bird judgments. Our alternative successful approach is to use both data sources simultaneously in the main training phase of deep learning.

Zebra finch syllables are complex and variable. Within syllables, they vary from millisecond to millisecond in acoustic features including pitch, harmonicity and peak frequency (see S4 Fig for examples). When zebra finches are faced with a task of comparing two syllables, such as in our experiment or in their natural behaviour, they must integrate these differences. Our ML algorithm was trained on these decisions and it is therefore not surprising that the simple summary statistics of these features, such as means, only partially capture its embedding, together accounting for only 68% of variation. Moreover the relative contributions of the different song features are similar to the parameter weight settings that resulted from tuning Luscinia with the birds' decisions (S1 Table) in that fundamental frequency was found to be the most informative single feature. Although this wasn't the primary goal of our study, these two sources of data also provide an indication of how birds integrate different perceptual features when assessing songs. It appears that mean fundamental frequency plays a particularly important role to zebra finches and that other temporal aspects of the pitch trajectory are, perhaps surprisingly, less critical. Nevertheless our results suggest that our comparison method, and perhaps the perception of birds is highly multidimensional.

The two algorithms that we trained with birds' judgments (Luscinia-U and our new machine learning algorithm Emb-LUA) outperformed other methods of song comparison on our testing data-set. This supports the idea that data-informed methods of comparison may be more reliable than ones that rely solely on human intuition. Of the remaining algorithms that we tested, the untrained version of Luscinia's dynamic time warping algorithm outperformed Sound Analysis Pro and the implementation of SPCC in Raven. Aspects of our study system and task may go some way to explaining these differences. Zebra finch song is highly spectrally complex compared to many birds' song. SPCC was originally employed on swamp sparrow song, where all spectral energy is concentrated at one frequency at any one time point, and it would seem logical that it might be harder to interpret spectrographic overlap in more complex scenarios. The algorithm used by Sound Analysis Pro was similarly designed for a particular task: to assess similarity between shared song syllables—i.e. pairs of syllables that share some degree of similarity. It was not designed to measure similarity between all pairs of notes, and beyond a certain level of dissimilarity, a floor of 0 is reached. This directly impacted its performance in our test: when removing affected triplets, its performance did not exceed that of Luscinia and our machine learning algorithm, but the three were relatively close in their performance. Most research on zebra finch song over the last two decades has used Sound Analysis Pro and our results would suggest that for tasks involving the assessment of the precision of song learning among relatively similar groups of syllables, these analyses have a reasonable degree of validity.

Although the algorithm we introduced here outperforms other commonly used methods and presents a major step forward in the measurement of song similarity, some care has to be taken. In this experiment we have presented birds with single song syllables. Although these isolated syllables, rather than full song, may have sounded less natural to the birds, and also limited the possibilities of measuring higher levels of song perceptions, they made it possible

to study sound perception using sounds with features they are naturally attuned to. Furthermore, although all syllables used originated from a single population (different from the experimental population), since the variation in syllables within a zebra finch colony is large compared to that between colonies [37] (but see [38]), we do not expect that this limits the generalisability of the algorithm for different zebra finch colonies. Finally, the current algorithm is only based on zebra finch perceptual judgements which may not be the same for all bird species. Different species have different hearing sensitivities, and may differ in how they weight different sound features [39–41]. Collecting similar perceptual judgements from a range of other species will enable us to compare differences in perception and will improve our overall ability to accurately measure song similarity.

## Supporting information

**S1 Fig. Operant feeder.** For a schematic image see Fig 1.
(TIFF)

**S2 Fig. Training stimuli and probes projected in 2-dimensional space.** Extracted from 64-dimensional embedding using T-SNE for the projection with Euclidean distances. Sets of training stimuli are connected with blue lines, connections between probes and training stimuli with grey lines. Stimuli used for training on the left side are coloured in red, training stimuli on the right side are coloured in green.
(TIFF)

**S3 Fig. Overview of deep learning model architecture.** Input is a time-frequency representation of a recording, with shape 170 time frames and 150 frequency bins. The shared convolutional part performs a number of convolutions, batch normalisations, and leaky-ReLU nonlinearities to the input. The output of the convolutional part is used as input to two different branches of the network, one performing attention pooling and the other performing max pooling. The results of the two branches are concatenated together and used as input to the final layer of the network; a dense layer that performs projection of the input into an N-dimensional embedding space. Figure adapted from [20].
(TIFF)

**S4 Fig. Mel spectrograms of a stimulus pair and 4 probes assessed by the bird trained with the stimulus pair.** Probes in the left column were assessed by the bird as being more similar to the left stimulus, and probes in the right column were assessed as being more similar to the right stimulus. Numbers indicate the dissimilarity between the probe and the left (L) and right (R) stimulus according to Luscinia. Note that for 3 probes the bird agrees with the assessment of Luscinia, while for one probe (M2210_4) it disagrees with the Luscinia assessment.
(TIFF)

**S5 Fig. Accuracy (%) of deep learning models based on the learnt embedding space dimensions.**
(TIFF)

**S6 Fig. Accuracy (%) results on the evaluation set after removing triplets that produce zero 'similarity' values in Sound Analysis Pro for both pairs in a triplet.** Details are the same as in Fig 5.
(TIFF)

**S7 Fig. Outcome of MRM analyses.** Results show the standardised regression coefficients (A) when grouping the variables by the 5 overall song features and syllable length and (B) when

grouping the variables by summary statistics start, end and mean. Each bar represents the standardised regression coefficient from the respective MRM analysis.
(TIFF)

**S8 Fig. Within and between stimulus set similarity.** Comparison of similarity within a stimulus category (close: core stimulus exemplar in group 'A' with the 7 other syllables of group A and the same for group B) and between stimulus categories (far: between core stimulus exemplar A and the 8 stimuli in group B, and vice versa) for Luscinia, SAP and Raven (SPCC) for the evaluation set. For all three algorithms the close and far groups separate out, showing a higher similarity for the close group than for the far group.
(TIFF)

**S1 Table. Table of parameter settings for Luscinia.**
(PDF)

**S2 Table. Table of MRM standardised regression coefficients.**
(PDF)

## Acknowledgments

We thank Neeltje Boogert for making zebra finch recordings available.

## Author Contributions

**Formal analysis:** Lies Zandberg, Veronica Morfi, Dan Stowell, Robert F. Lachlan.

**Investigation:** Lies Zandberg, Veronica Morfi, Julia M. George, Dan Stowell, Robert F. Lachlan.

**Methodology:** Lies Zandberg, Veronica Morfi, David F. Clayton, Dan Stowell, Robert F. Lachlan.

**Writing – original draft:** Lies Zandberg, Veronica Morfi, Dan Stowell, Robert F. Lachlan.

**Writing – review & editing:** Lies Zandberg, Veronica Morfi, Julia M. George, David F. Clayton, Dan Stowell, Robert F. Lachlan.

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
