## [Decision Letter · Decision Letter 0]

27 Nov 2023

Dear Dr. Zandberg,

Thank you very much for submitting your manuscript "Bird song comparison using deep learning trained from avian perceptual judgments" for consideration at PLOS Computational Biology.

As with all papers reviewed by the journal, your manuscript was reviewed by members of the editorial board and by several independent reviewers. In light of the reviews (below this email), we would like to invite the resubmission of a significantly-revised version that takes into account the reviewers' comments.

The manuscript "Bird song comparison using deep learning trained from avian perceptual judgments" presents novel research data and a novel deep learning based algorithm for bird song analysis. The manuscript is very well structured and written. Following the advice of two reviewers I recommend a major revision. Specially the limitations and potential bias introdiuced by the procedure used to perform data collection, as pointed out by the reviwers, need to be addresed.

We cannot make any decision about publication until we have seen the revised manuscript and your response to the reviewers' comments. Your revised manuscript is also likely to be sent to reviewers for further evaluation.

Sincerely,

Waldo Nogueira

Guest Editor

PLOS Computational Biology

Zhaolei Zhang

Section Editor

PLOS Computational Biology

The manuscript "Bird song comparison using deep learning trained from avian perceptual judgments" presents novel research data and a novel deep learning based algorithm for bird song analysis. The manuscript is very well structured and written. Following the advice of two reviewers I recommend a major revision. Specially the limitations and potential bias introdiuced by the procedure used to perform data collection, as pointed out by the reviwers, need to be addresed.

Reviewer's Responses to Questions

**Comments to the Authors:**

Reviewer #1: Review uploaded as an attachment

Reviewer #2: Bird song learning is a major field of study both in behavioral biology and neurosience, in which it is used as a neural model system for human speech development (rodents and primates don't vocally imitate). Most studies depend on quantifying the level of learning that took place under experimental or natural conditions, in order to study how learning takes place. The level of resemblance between original song (tutor) and copy (tutee) is either scored by human observers or estimated by relatively simple algorithms. These algorithms use spectrographic features as input that human observers consider salient, and are thus indirectly also based on human judgement/iterpretation. By a combining two novel appoaches, the current study constitutes an qualitative advancement to this state of affairs, which has been the norm for decades.

First, the authors base the reference for similarity not on direct or indirect human judgement, but rather on the species own perceptual judgements. This makes a lot of sense because a tutee mimics a tutor's song through its own perceptual systems, and the level of success should be based on that. The authors introduce an innovative way to achieve this, namely testing a comparatively large number of birds psychoacoustically in an operant setup, while remaining in their normal social group setting. Traditionally, birds need to be isolated for such tests, which less ideal for social birds, as it may lead to stress and/or depression, which one should avoid as much as possible for ethical reasons, but which may also lead to inferior results. The results in the current study show that with the developed paradigm, it is possible to test comparatively large numbers of birds in a social situation. This will have important consequences as the the standard model species for speech development in neuroscience is the zebra finch, which is highly social. Until now most studies do not attempt to measure bird judgements, and if they do this is based on a very low number of subjects. Further, the authors show that the most commonly used similarity algorithms (human judgement based), do not closely predict bird perceptual judgement, although they do perform better than chance.

Second, the authors developed a deep-learning algorithm to predict the bird perceptual judgments. This algorithm outperforms both classic human judgement-based algorithms, as well a variant thereof that is weighted using the bird perceptual data. This shows that the novel deep-learning algorithm successfully includes features in the input data that are not captured with traditional techniques.

Taken together these are valuable results that undoubtedly will form the basis for better quantification of vocal similarity in many future studies.

However, I have some concerns and remaining questions after reading the manuscript:

1) How was it monitored that the birds received enough food? The only way the birds could get food is through the operant device (l.94). How was it verified that they were sufficiently successful? Presumably 50% correct is also a way to get food (since there is no real punishment), but it would be good to see that this was monitored in some way.

2) The text is sloppy here and there in suggesting the neural network models the birds' sound perception (l. 247). The authors probably do not intend to literally claim this, but should be more precise. The model mimics the birds' categorization behavior based on the same sound input, but it could use correlated features.

3) A bit related to this: what is the actual input to the artifical neural network? The text mentions 'audio', which implies a sampled sound pressure waveform, but in l. 222 it is mentioned that recordings are first transformend into power mel spectrograms. This may be relevant because the transformation will perhaps filter out information that is normally available and used by neural auditory systems, such as phase. From the results it is clear that the algorithm does well without it, but it is relevant because such information could be a target for future improvements.

4) I can find little information on the likelihood of observational learning playing a role in the experiment. The birds can hear and see a bird respond in the operant aparatus I presume. Is this expected to have played a role?

Other:

l 101: Two speakers: why two and does this have an influence on the results? Design 1 had one speaker.

l. 119: "In quick succession". With an interstimulus interval? If so how long? Was the input to the algorithms also repeated? The end of one syllable and the start of the next one could be a 'feature' that is used by the birds.

l. 165: Why is the criterion to move on to the next stage based on group performance? Why not per individual bird?

l. 222: 170 time frames. The syllables had different durations I presume. Does that mean that the temporal resolution of time was different between syllables?

l. 247: "model the way birds perceive sounds". I do not think this is necessarily what happens. The system categorizes zebra finch syllables like zebra finches do.

l. 347: "syllable length", should perhaps be "syllable duration"?

**Have the authors made all data and (if applicable) computational code underlying the findings in their manuscript fully available?**

Reviewer #1: Yes

Reviewer #2: Yes

PLOS authors have the option to publish the peer review history of their article (what does this mean?). If published, this will include your full peer review and any attached files.

Reviewer #1: **Yes: **Julie E Elie

Reviewer #2: No
---

## [Decision Letter · Decision Letter 1]

19 May 2024

Dear Dr. Zandberg,

Thank you very much for submitting your manuscript "Bird song comparison using deep learning trained from avian perceptual judgments" for consideration at PLOS Computational Biology. As with all papers reviewed by the journal, your manuscript was reviewed by members of the editorial board and by several independent reviewers. The reviewers appreciated the attention to an important topic. Based on the reviews, we are likely to accept this manuscript for publication, providing that you modify the manuscript according to the review recommendations.

The authors have significantly improved the manuscript and have addressed most comments by the two reviewers. Still, one of the reviewers has minor but important comments that need to be adressed before accepting the manuscript for its publication.

Sincerely,

Waldo Nogueira

Guest Editor

PLOS Computational Biology

Zhaolei Zhang

Section Editor

PLOS Computational Biology

The authors have significantly improved the manuscript and have addressed most comments by the two reviewers. Still, one of the reviewers has minor but important comments that need to be adressed before accepting the manuscript for its publication.

Reviewer's Responses to Questions

**Comments to the Authors:**

Reviewer #1: With this revision, the authors have substantially increase their article clarity. Figure 2 is much more clear that the previous version. Overall the paper is really improved. My general comments are the following:

1- I think it is worth discussing the potential bias in the behavioral classification of syllables due to the choice of the training sets based on Luscinia's distance calculation. I appreciate the figure of pair distances (close/far) that the authors provided in their response, and understand that gathering new behavioral data is not reasonable, but I think it is important to discuss this aspect in the paper, and comment on how the change of distance metric could have affected the syllables ranking and choice of training sets. In the end, training a cohort of birds might not be much different than training an algorithm: whatever bias is present in the training set shows up in the algorithm/birds decisions.

2- I think it would be useful to provide some contexts to, and work on the wordings of, the result sections line 434-448 (I tried to give some indications of what is unsettling below in my line by line comments).

3- Finally, it would worth it to investigate a bit more thoroughly the embedding obtained by the model (Song Feature section, which by the way, I would suggest to change the title of to something like: acoustic features used by the algorithm) and in particular increase the number of bioacoustic measurements used in the analysis. Interpreting the acoustic dimensions that the model is using to emulate the birds responses is valuable for understanding what birds might mostly value when evaluating song syllable similarity. Do you think birds might have adapted their perceptual judgement to the proposed set of training stimuli (weighting in more the acoustic features that were the most distinct between set A and set B of syllables) or are these results actually showing that they kept attending to features (such as the fundamental frequency) that were not necessarily the best to discriminate the two sets of syllables? Then it becomes interesting to see the importance of the fundamental frequency in emulating zebra finches' choices.

Line by line comments:

L186-187 : Make clear that both stimuli are rewarded

Line 221: double point

L241: double parenthesis

L278-280: to be consistent, does a probe decision has to be the same over at least 2 decisions OR 70% of the times the probe was played to the birds? These two conditions do not seem to have the same contingencies to me, but maybe because on average a probe is presented 2-3 times, that ends up being 70%?. Also if it is a single individual response, why “played to THEM”?

L326: Please add if I am correct, if not I am not sure to understand how the cycle accuracy is calculated: correct responses among all “training” responses to stimuli during the test phases.

L328: what is inter-rater here? Do you mean inter-bird?

L330-332: this sentence is awkward to me. I don’t quite make sense of the last part: “as estimated maximum accuracy (consistency assumption)”. Something seems to be missing.

L353: do you mean “with flat priors across which parameter weights should be kept or updated in the DTW algorithm”?

L352-355: Are they indeed 2 steps to decide of the weights (Metropolis-Hastings rule + MCMC) or is there some sort of repetition here that makes the text unclear?

L393-395: can the authors indicate the number of different probe syllables for which data where obtained?

L432: revise something like: “We added all software methods to the MDS defined 2-d space that enables the comparison of model output.”

L434: rephrase “when evaluating systems”

L434-435: this beginning of paragraph is unsettling: the reader has to guess what we are now looking at. “both pairs of similarities….” = algorithm similarity? Probe-training sound distances? Systems = algorithms/software methods?

L441 The transition to this paragraph is really dry. Consider adding some more context.

L442: According to methods L375-382 only 6 parameters are tested for their power to predict the distance matrix obtained with EmbLUA, where is the 26 acoustic features coming from?

L445: I understand from these results that the embedding of EmbLUA can be explained at 56% by the differences in fundamental frequencies between song syllables, and then at 60.6-56 = 4.6% by the other 5 parameters tested, which leaves around 40% for other criteria. One parameter explaining 56% of the decisions criteria of Luscinia is not quite what I would name a comprehensive set of acoustic features… Maybe I am missing the point here, but the authors should revise the phrasing in this paragraph.

FigS7: Please revise the legend. It is very difficult to make sense of it as is. E.g.: can the author specify here which model was tested (I presume EmbLUA?), what beta corresponds to on the yaxis (I guess weights of the regression model?). I can’t make sense of the legend sentence (“dissimilarity for the measures taken from fundamental frequency”?)

L467-469: what would be the alternative? This task is a force choice test based on acoustic only, so I am not sure to understand what the authors mean here.

L472-475: this sentence is hard to read, consider rephrasing in shorter sentences.

471-472: The assumption made by the algorithm is also that the log mel spectrogram is a good representation of the zebra finch song syllables, which is not necessarily the case! When comparing the accuracy of the same classifier using different feature spaces (ad hoc bioacoustics features vs spectrogram vs mel scepstrum) at classifying zebra finches vocalization types, the mel space was found to be a poor feature space in Elie and Theunissen 2016. So it is possible that in addition to the explanations that the authors already mention, the input to the algorithm itself is not optimal. Could the author add examples of the actual input to the deep learning model ? Fig S4 is great in providing examples in the spectrogram space, but I would be curious to see the corresponding log mel spectrograms.

484-492: The authors make a strong statement that the birds and EmbLUA make use of spectro-temporal features that are not well represented by traditional bioacoustics measurements when they tested only 6 measures and one of them (fundamental frequency) actually explains alone more than half of EmbLUA embedding distances between syllables. I think this statement should be tuned down or results with a more comprehensive set of parameters should be provided.

Besides, this analysis is very interesting, because it opens a window into what acoustic features might be used by the birds. To corroborate/illustrate the importance/usefulness for the birds of these acoustic features, one could look at the difference of them between unambiguous probe-training pairs of syllables and ambiguous probe-training pairs of syllables.

493: what is “our training dataset”? Name what you refer to with “new machine learning algorithm”? Is that EmbLUA? Since different datasets (bird generated, Luscinia generated) and different algorithms are presented in this paper, this sentence is a bit confusing.

Reviewer #2: I am happy with the revision. All my concerns with the previous version have been addressed well.

**Have the authors made all data and (if applicable) computational code underlying the findings in their manuscript fully available?**

Reviewer #1: None

Reviewer #2: Yes

PLOS authors have the option to publish the peer review history of their article (what does this mean?). If published, this will include your full peer review and any attached files.

Reviewer #1: **Yes: **Julie E Elie

Reviewer #2: No

Figure Files:

Data Requirements:

Reproducibility:

References:

---

## [Decision Letter · Decision Letter 2]

15 Jul 2024

Dear Dr. Zandberg,

We are pleased to inform you that your manuscript 'Bird song comparison using deep learning trained from avian perceptual judgments' has been provisionally accepted for publication in PLOS Computational Biology.

Best regards,

Waldo Nogueira

Academic Editor

PLOS Computational Biology

Zhaolei Zhang

Section Editor

PLOS Computational Biology

Thank you very much to the authors for addressing all the comments. The manuscript is ready for publication. Please consider the minor comments from reviewer 1.

Reviewer's Responses to Questions

**Comments to the Authors:**

Reviewer #1: Reading again the entire article, I think this study will be a great addition to the literature, in particular with the published code and dataset. Congratulations to the authors! I have 5 minor points that I think should be addressed before publication.

Line 241-242: The mel spectrogram is tuned for human hearing and it remains to be shown if zebra finch perception is as close to human sound perception as is implied by the authors. Please note that others have directly compared different feature spaces and found other feature spaces to be more informative about acoustic differences in the zebra finch repertoire (see for instance Elie and Theunissen, Animal Cognition, 2016). As far as I can tell, Morfi et al did not compare between different sound representations, but cite Sinnott et al 1980 (a psychoacoustic study in pigeons, blackbirds and cowbirds that show some similarity but also some differences to human perception) to justify their choice.

Figure 2 : can the author specify how the data binning was performed and overlay a density plot for the values obtained for the 753 probes? As such it would be impossible to reproduce the figure and it is difficult to evaluate how the binning is representative of the entire dataset.

Figure S3 could benefit from some legend in particular define what “dense” and “concat” mean. Since this figure is taken from Morfi et al 2021, you should use the legend of that paper and indicate that this figure is coming from Morfi et al 2021.

In order to illustrate the acoustic features of Luscinia used in the paper, can the author add a supplementary figure of Mel spectrograms of syllable examples with overlayed results of acoustic feature calculations? It would be useful to have this in mind when reading discussion line 523 onwards and would enhance the understanding of the methods.

Fig S8 Maybe indicate Raven for the cross-correlation for consistency with previous figures?

**Have the authors made all data and (if applicable) computational code underlying the findings in their manuscript fully available?**

Reviewer #1: Yes

PLOS authors have the option to publish the peer review history of their article (what does this mean?). If published, this will include your full peer review and any attached files.

Reviewer #1: **Yes: **Julie E Elie

---

## [Editor Report · Acceptance letter]

2 Aug 2024

PCOMPBIOL-D-23-01544R2 

Bird song comparison using deep learning trained from avian perceptual judgments

Dear Dr Zandberg,

I am pleased to inform you that your manuscript has been formally accepted for publication in PLOS Computational Biology. Your manuscript is now with our production department and you will be notified of the publication date in due course.

With kind regards,

Anita Estes
